# Identifying HIV-1 Transmission Clusters in Uzbekistan through Analysis of Molecular Surveillance Data

**DOI:** 10.3390/v14081675

**Published:** 2022-07-29

**Authors:** Aleksey Lebedev, Anna Kuznetsova, Kristina Kim, Ekaterina Ozhmegova, Anastasiia Antonova, Elena Kazennova, Aleksandr Tumanov, Adkhamjon Mamatkulov, Evgeniya Kazakova, Nargiz Ibadullaeva, Krestina Brigida, Erkin Musabaev, Dildora Mustafaeva, Visola Rakhimova, Marina Bobkova

**Affiliations:** 1Gamaleya National Research Center for Epidemiology and Microbiology, 123098 Moscow, Russia; a-myznikova@list.ru (A.K.); kimsya99@gmail.com (K.K.); belokopytova.01@mail.ru (E.O.); anastaseika95@mail.ru (A.A.); kazennova@rambler.ru (E.K.); desep@mail.ru (A.T.); mrbobkova@mail.ru (M.B.); 2Research Institute of Virology, Tashkent 100194, Uzbekistan; amamatkul@gmail.com (A.M.); dr.kazakova.evg@gmail.com (E.K.); inargis@bk.ru (N.I.); dr.kristina@mail.ru (K.B.); drmusabaev1956@gmail.com (E.M.); 3Republican AIDS Center, The Ministry of Health, Tashkent 100135, Uzbekistan; mustafaeva_1969@list.ru; 4Center for Development of Profession Qualification of Medical Workers, Tashkent 100007, Uzbekistan; visolasitora@mail.ru

**Keywords:** HIV-1, subtypes, molecular epidemiology, transmission clusters, migration patterns, Uzbekistan

## Abstract

The CRF02_AG and sub-subtype A6 are currently the predominant HIV-1 variants in the Republic of Uzbekistan, but little is known about their time-spatial clustering patterns in high-risk populations. We have applied molecular evolution methods and network analyses to better understand the transmission patterns of these subtypes by analyzing 316 *pol* sequences obtained during the surveillance study of HIV drug resistance. Network analysis showed that about one third of the HIV infected persons were organized into clusters, including large clusters with more than 35 members. These clusters were composed mostly of injecting drug users and/or heterosexuals, with women having mainly high centrality within networks identified in both subtypes. Phylogenetic analyses of the ‘Uzbek’ sequences, including those publicly available, show that Russia and Ukraine played a role as the main sources of the current subtype A6 epidemic in the Republic. At the same time, Uzbekistan has been a local center of the CRF02_AG epidemic spread in the former USSR since the early 2000s. Both of these HIV-1 variants continue to spread in Uzbekistan, highlighting the importance of identifying transmission networks and transmission clusters to prevent further HIV spread, and the need for HIV prevention and education campaigns in high-risk groups.

## 1. Introduction

Thirty years ago, in 1991, the Soviet Union broke up and its 15 republics gained independence. After the initial socio-economic turmoil, most of them started to recover, but they are still facing the enormous challenge of controlling the rapid spread of HIV infection, which reached epidemic proportions. Approximately 2.2 million people in Eastern Europe and Central Asia were living with HIV in 2020; about 58,000 of these were in the Republic of Uzbekistan [1,2]. During 2020, 3983 new HIV infections were diagnosed in the Republic. Of these new infections, 73.2% were associated with sexual transmission, which is considered to be the most important factor driving HIV spread in Uzbekistan [3]. Nevertheless, around 18.2% of new infections occur in injecting drug users (IDUs) [3], with HIV prevalence in this group reaching nearly 30.0% [4,5]. In Uzbekistan and many other Central Asian countries, sexual risks—primarily heterosexual (HET)—become conflated with risks associated with injection drug usage [6,7,8]. At least 20% of female sex workers (FSW) are believed to inject drugs, which increase the risk of infection by 5–6 times [6,7]. The prevalence of HIV infection is high among Uzbek migrants (4.1%) [4], who are regarded as a driving force of the HIV-1 spread in former Soviet Union (FSU) countries [9,10,11]. Given all this, the identification of transmission networks and clusters affecting key populations is of paramount importance for preventing further HIV transmission.

The HIV-1 subtype patterns in Uzbekistan are complex and continuously changing in response to human population migration and active transmission networks. In the Republic, as many as six known HIV-1 subtypes and circulating recombinant forms (CRFs) have been detected, with CRF02_AG (57.0%) and sub-subtype A6 (41.0%) accounting for 98% of total infections [12]. The latest National HIV Molecular Epidemiological Survey performed by our group revealed that CRF02_AG, initially identified in IDUs in the capital [13], has quickly overtaken subtype A6 in this vulnerable population during the past few years. Furthermore, CRF02_AG also became dominant among HETs, increasing from 6.0% in 2003 to 53.0% in 2016 [12,14]. According to our estimates, CRF02_AG today can account for 60–65% of all HIV-infections in Uzbekistan and is likely to increase further. For this reason, it is imperative to understand the trends of CRF02_AG and A6 epidemics in Uzbekistan in order to inform targeted public health interventions. 

The aim of our current study was to examine phylogenetic clustering, cluster composition and factors associated with clustering in a cohort of ‘Uzbek’ HIV+ people infected with two dominating HIV-1 subtypes: CRF02_AG and A6.

## 2. Materials and Methods

### 2.1. Sequence Dataset Compilation

HIV-1 sequences included in this study were derived mostly (95.6%) from participants in the Nationwide Surveillance of HIV Drug Resistance (NS-HIVDR) program [12], which constitutes a well-defined and demographically diverse cohort of antiretroviral therapy (ART)-naive and ART-experienced HIV-positive people from 13 different Uzbekistan regions. A total of 316 HIV-1 sequences covering the complete protease (PR) and a part of the reverse transcriptase (RT) of *pol* (polymerase) gene (nucleotides 2253 to 3344 of reference strain HXB2; 1092 bp long) were used, of which 59.8% (188/316) were sampled from participants infected with HIV-1 CRF02_AG, and 40.2% (127/316) from patients infected with HIV-1 subtype A6. 

The blood plasma samples were collected during 2010–2016 at the Research Institute of Virology of the Ministry of Health of the Republic of Uzbekistan, or at local hospitals in different regions of the Republic; in addition, 37 sequences (unpublished studies) were retrieved from the Los Alamos HIV Sequence Database (https://www.hiv.lanl.gov/content/index, accessed on 20 May 2021) (Figure 1; Appendix A). The epidemiological metadata, including transmission route, age, sex, country of infection, sampling area, immunological (CD4+ T-cell count) and virological (viral load) information, were available for 210 of 316 sequences and used as characteristic data. A total of 739 CRF02_AG/CRF063 (N = 346) and subtype A6 (N = 393) *pol* sequences most closely related to each Uzbekistan sequence from the BLAST search at GenBank (with >95% sequence identity and de-duplicated) were also used; in particular, the CRF063_02A6 sequences were used to improve the clock-like structure in datasets, but were not used to reconstruct migration pathways. The GenBank accession numbers for sequences used in this study are provided as Appendix A.

### 2.2. Sequence Alignment and Subtype Assignment

Nucleotide sequences were aligned using MAFFT online service (https://mafft.cbrc.jp/alignment/server/, accessed on 20 May 2021). HIV-1 subtype analysis was performed by our group earlier and described in the article by Mamatkulov et al. [12]. The RIP tools (https://www.hiv.lanl.gov/content/sequence/RIP/RIP.html, accessed on 20 May 2021) were also used for recombination analysis. 

### 2.3. Identification of Transmission Clusters and Uzbek-Origin Clusters

The clusters of closely related sequences (transmission clusters) and potential origin of the HIV-1 clades were evaluated by phylogenetic approach, as described in the study by Alexiev et al. [15]. Briefly, we used the MicrobeTrace tool [16] to cluster selections in ML trees using the sequence alignment, including all 316 ‘Uzbek’ *pol* sequences, at Tamura–Nei pairwise genetic distance (d) threshold of 0.015 substitutions/site and having at least 3 sequences; the ML trees in Newick format were importing into MicrobeTrace after the reconstruction by Nextstrain tool (https://nextstrain.org/, accessed on 20 May 2021) under GTR (general time reversible) model. Visualizing clusters by node and total number of links in the transmission network was performed in MicrobeTrace. The degree of centrality, measured as the number of direct connections a node has to other nodes (that is, how many direct “one hop” connections each node has to other nodes in the network), was manually estimated based on the formula contained in [17]. We then sought epidemiological predictors of clustering and calculated assortativity for epidemiological variables such as region, sex, and transmission category.

To evaluate the potential origins of HIV-1 clusters in Uzbekistan, all sequences were combined with those from other countries and analyzed with the Nextstrain tool (https://nextstrain.org/) that uses ML analysis implemented in Tree-Time [18,19]. The total dataset for Nextstrain ML analysis contained 1055 HIV-1 sequences, comprising 316 sequences from Uzbekistan and 739 global HIV-1 *pol* sequences, plus one HIV-1 subtype C sequence (GenBank accession numbers: AF443091) used as the outgroup. The output JASON files from Nextstrain pipeline were viewed using Auspice (https://auspice.us, accessed on 20 May 2021).

### 2.4. Statistical Analysis

Epidemiological data, including sex, age, country of birth, region of Uzbekistan, and transmission categories, were analyzed by Pearson’s chi-squared or two-tailed Fisher’s exact test (where necessary). Multivariate logistic regression analysis was performed with the outcome variable as HIV-1 subtype: CRF02_AG or A6. Odds ratios were estimated with 95% confidence intervals (CIs). The differences were considered significant at a *p*-value less than 0.05. All analyses were performed in STATISTICA v.10.0 software (StatSoft, Tulsa, OK, USA). Assortativity coefficients for selected variables were calculated using the Python v3.9.5 (https://www.python.org/downloads/release/python-395/, accessed on 20 May 2021) script by Sergey-Knyazev (https://github.com/Sergey-Knyazev/attribute_assortativity, accessed on 20 May 2021) and NetworkX package (https://networkx.org, accessed on 20 May 2021), with thresholds of d = 0.015 substitutions/site (1.5%).

## 3. Results

### 3.1. Association of Demographic Characteristics with HIV-1 Infection in Uzbekistan

To perform multivariable analysis, we used a subset of 210 HIV-positive people for whom there was information about all the variables considered: sex, risk group, region of residence, and age. Of these, 124 (59.0%) sequences belonged to CRF02_AG and 86 (41.0%) sequences belonged to sub-subtype A6 HIV-1. It was also decided to combine the sequences from the north-eastern part of the Republic (Andijan, Fergana, Namangan and Tashkent regions) into a single group termed ‘North Eastern Region’ (NER). The multivariable analysis found an association between the region of residence and HIV-1 subtype (Table 1). So, HIV-positive people from NER were more likely to have the CRF02_AG variant compared to those infected in other regions (OR = 2.62, 95% CI = 1.44–4.75, *p* = 0.001). Moreover, this analysis produced the same results as when using all sequences from Uzbekistan (N = 316); CRF02_AG was still the most prevalent HIV variant in NER, although the proportion (137/316, 43.4%) was higher than in the subset of 210 people (80/210, 38.1%) (Appendix A). There was no evidence of association of other characteristics with HIV-1 subtypes.

### 3.2. HIV-1 Subtype Transmission Clusters in Uzbekistan

The phylogeny of HIV-1 CRF02_AG (N = 189) and A6 strains (N = 127), followed by processing and analysis in MicrobeTrace, indicated the presence of four transmission clusters, including two large clusters containing over 35 members, one for each subtype (Figure 2; Table 2). In total, 28.8% (91/316) of people from the dataset were members of transmission clusters. The largest cluster (CRF02_AG; N = 49) mainly comprised women (67.3%) and IDUs (38.8%), with a majority of people living in Andijan (N = 25, 51.0%), Tashkent (N = 7, 14.3%), Fergana (N = 4, 8.2%), and Samarkand (N = 8, 16.3%) regions; that is, about 90% of cluster members lived in the NER. By contrast, the second largest cluster (sub-subtype A6; N = 36) consisted mostly of men (52.7%) and HETs (44.0%); about two-thirds of them were from Tashkent (N = 22, 61.1%), while the rest were mainly from Andijan (N = 4, 11.1%) and the Samarkand region (N = 3, 8.3%), i.e., more than 72% were from the NER. The members from remaining clusters (N = 3, in each cluster) were mostly HETs from a single home region (such as Xorazm). It is interesting to note that women had a significant presence within networks identified in both the CRF02_AG and A6 clusters. With regard to the characteristics of people involved in the clusters, all were natives of Uzbekistan; the overall proportion of labor migrants among people in our study could be as much as 30% (personal data).

We also calculated assortativity among HIV-infected people in major clusters and dyads by sex, region, and transmission category using assortativity (r) coefficients (Table 3). The r-value characterizes the degree of mixing (i.e., disassortativity) or non-mixing (i.e., assortativity) of characteristics between entities that are connected within a network. The r-value ranges from −1 to 1, where negative values represent dissasortativity or mixing between entities of different characteristics, and positive values represent assortativity or non-mixing; zero value represents random mixing with no tendency for assortativity [20]. Our analysis found that region was the only characteristic with high assortativity (r > 0.25), being variable among HIV subtypes and highest among CRF02_AG (r = 0.30; total, node size = 67) (Table 3). The set was modestly dissasortative (r > 0.25) by HIV-transmission risk for HIV-1 subtype A6-infected people.

### 3.3. Origin and Cross(in)-Country Transmission of HIV-1 Sub-Subtype A6 and CRF02_AG in Uzbekistan

Here, we combined NS-HIVDR subtype A6 and CRF02_AG sequences with similar sequences of the same subtypes identified by BLAST search in HIV Databases at LANL (Los-Alamos National Laboratory), followed by data analysis in Nextstrain. A visual inspection of our phylogenetic trees suggested a certain degree of genetic structure based on geographical location (Figure 3). Analysis of CRF02_AG viruses shows that the majority of transmission events took place in Central Asia and formed a large older clade, with ancestor root location most probably traced to Uzbekistan. The CRF02_AG strains from Uzbekistan, Kyrgyzstan, Tajikistan and Kazakhstan, and a small number from Russia, are intermingled within this clade to form several minor country-specific subclades; overall, apart from one subclade formed by the Russian CRF063_026A sequences (marked with an asterisk), no other country showed a clearly compartmentalized grouping of sequences (Figure 3A). The only more or less large ‘Uzbek’ subclade (scl-1, N = 94) consisted of 30.2% (59/189) of ‘Uzbek’ sequences from people living mostly in the Andijan (50.9%, 29/57) and Tashkent (24.6%, 14/57) regions and included all sequences of a large transmission cluster, described above; the overwhelming majority of them were related to IDUs (36.8%, 21/57). The minor Uzbek subclade 2 (scl-2, N = 36) and subclade 3 (scl-3, N = 27) included of 77.8% (28/36) and 66.6% (18/27) Uzbek sequences, respectively; most peoples in both subclade (≥50%) were from Tashkent and with a comparable proportion IDUs and HET (~30%).

The finding that almost all ‘Uzbek’ CRF02_AG sequences could be traced back to a unique most recent common ancestor (MRCA) suggests a single major introduction event, followed by its local spread. Using an evolutionary time scale, the Nextstrain analysis indicated that the CRF02_AG outbreak cluster in Uzbekistan evolved from a MRCA introduced around 25-05-2000 [12-02-1998, 14-04-2001]. The MRCA for one large and two minor ‘Uzbek’ subclades corresponded to 02-03-2003 [28-04-2002, 12-12-2004], 29-08-2001 [27-01-2001, 12-06-2002] and 27-01-2003 [10-03-2002, 21-08-2004], respectively. 

The reconstruction of migration pathways between different geographic areas showed that the early dissemination of CRF02_AG most likely occurred in Uzbekistan and Kazakhstan, and thereafter the epidemic spread to Kyrgyzstan and Russia (Figure 3C). We also found that the most significant migration routes of HIV-1 CRF02_AG showed directionality from Uzbekistan to these countries and only very few transmissions went in the opposite direction. Regarding the intra-country migrations events, there were 15 HIV transmission pathways found, of which 11 involved the city of Tashkent (Figure 4A).

Unlike CRF02_AG, the majority of sub-subtype A6 transmission events took place in East European countries (Belarus, Russia and Ukraine), followed by some regional isolation after local introduction (as seen for Uzbekistan) and spread and mixing in other geographic locations (as for infections in Russia) (Figure 3B). We revealed at least three clades consisting almost exclusively of ‘Uzbek’ sequences. The large clade (cl-1, N = 75) contained about half of all ‘Uzbek’ sequences (48.0%, 61/127) and included all sequences of the large transmission cluster described above. The majority of cases (60.7%, 37/61) in this clade were diagnosed in the capital city of Tashkent, with most of them 47.5% (29/61) being IDUs, followed by HETs (32.8%, 20/61). The medium-sized clade (cl-2, N = 14) contained 78.6% (11/14) Uzbek HIV cases from Samarkand (81.8%, 9/11) with an unknown risk of infection (54.5%, 6/11). The smallest clade (cl-3, N = 10) comprised 90% Uzbek HIV-positive people, mainly HETs (55.6%, 5/9) from six different regions. 

The most probable geographic origin of the large and smallest clades was Russia; there were ten A6 sequences from Ukraine near the root of medium-sized clade. Maximum-likelihood phylodynamic analysis inferred the MRCA [95% HPD] for large, medium-sized and smallest subclades at 08-02-2000 [11-02-1999, 31-07-2001], 09-08-2000 [29-12-1998, 29-01-2002] and 09-08-2000 [29-11-1998, 29-07-2002], respectively. The MRCA for the complete clade of 519 subtype A6 taxa corresponded to 03-12-1994 [95% HPD: 03-11-1991, 07-03-1996]. 

The analysis of migration pathways between different geographic areas supported at least five HIV-1 sub-subtype A6 migration events into Uzbekistan (Figure 3D). As expected, the original HIV-1 A6 source for at least some of the infections in Uzbekistan probably originated from Ukraine; the same was true for Russia and Belarus, which were affected by strong gene flow from Ukraine. However, the most significant migrations routes of HIV-1 subtype A6 showed directionality from Russia to Uzbekistan and back. We also found HIV transmissions from Uzbekistan to Tajikistan, Kazakhstan and Kyrgyzstan. Intra-country migration events (N = 12) mainly related to the city of Tashkent (N = 7) as in the previous case, and also Fergana/Samarkand (N = 3) (Figure 4B).

## 4. Discussion

The HIV epidemic has been going on for over 20 years in Uzbekistan, and transmission patterns have gradually evolved from injecting drug usage to heterosexual contacts [3,12,13]. Unlike FSU countries with a single HIV subtype (e.g., Caucasian republics with subtype A6 at more than 85.0% [21,22,23]), today there are two principal HIV-1 subtypes, CRF02_AG and A6 in about equal proportions, in Uzbekistan [12]. Our current data suggest that the majority of CRF02_AG infected people in Uzbekistan (43.4%) live in the NER, where the possibility of being infected with this HIV-1 variant and not A6 sub-subtype is twice as high as subject from other regions. The earlier dispersal of CRF02_AG to Tashkent and regions bordering the capital (Andijan, Fergana, Namangan and Syrdaryo), together with tight epidemiological links between these locations due to their geographical proximity and high level of people migration, as well as possible introduction of CRF02_AG into highly-connected transmission networks, appear to be the most likely causes of this phenomenon. In this study, we investigated the transmission networks of the most prevalent HIV subtypes by association of all available sequences with the demographic and epidemiologic data. 

To date, there is no consensus on methodology to infer HIV-1 molecular networks, but it is obvious that threshold selection is a key factor in network construction [24,25]. Here, we used single optimal thresholds (1.5%) for both subtypes. The network analysis showed that just over a third of HIV+ individuals were organized into clusters (including large clusters of over 35 members), the majority of which were clusters that contained non-dyads. Among people involved in clusters, all were natives of Uzbekistan, with a majority (over 70%) living in the NER (Andijan, Fergana, Namangan, and Tashkent regions). We found that women were slightly more likely to participate in transmission clusters, a finding that aligned with the current reports describing an increase in infections among women in Uzbekistan [3,12]. Moreover, these data highlight that females may be engaged in riskier sexual and drug-related behavior and they could be better supported by the expansion of molecular-based surveillance strategies to reduce transmission. 

It was previously established that 2–3 out of every 1000 women in Tashkent are involved in sex work [26]. In addition to this, about 20–24% of male IDUs have contacts with FSWs [27]. Female IDUs also exist and are highly susceptible to HIV infection through exchanging sex for drugs or money and risky injecting drug use, which increases the overlap between these risk groups [6,27,28]. All of this may explain the patterns of group risk mixing that we have identified in transmission clusters. Data in this report also indicate that IDU/HET-women, having high centrality within the highly connected network, can contribute more CRF02_AG infections than IDU-men, and a strategy to disrupt these contacts would be more effective in reducing infections than randomly targeted prevention approaches.

Although the high percentage of non-clustered sequences might be explained by low number of sequences included in the study or low sequence completeness (lack of sequences from HIV-infected persons with closely related viruses) [29], dead-end transmission events (non-transmitted infections) or infection abroad can also be seen as possible causes. The Uzbek labor migrants (or any other immigrants) are at an increased risk of HIV infection, since they often stay away from their hometowns for long time periods, and instead of having a monogamous sexual partner have multiple sexual contacts with FSWs. Although the high assortativity by regions within the country mostly supports local genetic connections, and the local populations are more likely to be infected inside the country (e.g., through regional networks), the migrant population needs urgent attention because HIV prevalence appears to be growing in this group.

Migration is one of many social factors contributing to the HIV/AIDS epidemic; the male/female migrants become infected while away from home and infect their spouses or regular partners upon return [30,31,32]. Uzbek migrants are mostly males going to Russia and Kazakhstan for work. They tend to have limited HIV knowledge and engage in sexual contacts involving sporadic use of condoms [33]. Our study supports this to be the case, but there is additional evidence for the bi-directionality of HIV transmission (between the country of birth to the country of residence): the results of phylogenetic study of CRF02_AG and A6 sub-subtypes in Uzbekistan are indirect indications (but not evidence) of this, in addition to the patient interviews. 

It has been previously shown that after leaving Africa, CRF02_AG initially extended to IDUs and grew rapidly in Uzbekistan, followed by dissemination across the FSU countries, including Kazakhstan and Russia [34,35,36]. According to the latest data, this expansion originated from a single viral clade that arose in 1996, and the subsequent epidemic growth occurred approximately during 1998–2003 [34,37]. Our results are in full accordance with these findings, since we estimated that the time to MRCA (TMRCA) of this and the other FSU countries’ epidemics to be around 1996, with the epicenter in Tashkent. 

In the late 1990s/early 2000s, Central Asian FSU countries were experiencing an explosive HIV-infection epidemic, which affected IDUs and their sexual partners [38,39,40,41,42]. To a large extent this was due to the location of the Central Asian drug trafficking routes (‘northern routes’) from Afghanistan towards their primary markets in Russia and Europe. As a result, 524 new HIV cases were diagnosed in Uzbekistan alone between 1999 and 2000 [14]. As with other IDU-associated HIV-infection epidemics in FSU during this period [43,44,45,46], the main mode of HIV transmission was syringe- and needle-sharing. These historical records support our phylogenetic findings. According to our data, CRF02_AG, after some regional isolation in Uzbekistan, as evidenced by several all-‘Uzbek’ clades, was introduced to the Central Asian countries and then Russia during a relatively short period via the IDU network; this may also explain the ‘similar’ HIV-1 epidemic on both sides of the Uzbekistan-Kazakhstan border (or Kyrgyzstan, and or Tajikistan [47,48]). Interestingly, we found only marginal cross-border CRF02_AG transmissions from FSU countries back to Uzbekistan, even in the case of Russia. Being a major hub for the Uzbek migrant populations, Russia is thus acting as a conductor zone, providing HIV transmission between the other FSU countries’. Even though in Russia other AG recombinants (CRF063) are widespread, CRF02_AG today only accounts for 3–5% of all HIV-infections in the country [49,50,51,52].

As part of the A6 epidemic, which was first described in the IDUs’ population, but spread among HET and even MSM (men who have sex with men) populations soon afterwards [7,8,9,10], we inferred at least two large ‘Uzbek’ A6 clades dated to the year 1999. The ancestral origin for these clades was traced with high probability to Russia and Ukraine, which supports previous findings [53] and reflects the primary role of these countries as the center of HIV epidemic spread to Uzbekistan during the late 1990s/early 2000s. Moreover, our findings also suggest the possibility of multiple introductions of A6 into Uzbekistan from these countries. However, we also report the dissemination of sub-subtype A6 from Uzbekistan back to Russia and Ukraine (as well as exportation to Tajikistan), with the participation of migrants infected through regional FSW-networks. We cannot rule out the possibility that migrant men were infected from their wives, who acquired the infection from their parallel sexual partners [33]. Overall, labor HET-migrants today may be a main factor fueling the HIV-1 bidirectional transmission between FSU countries. All of this points to the need for HIV prevention and education campaigns among high-risk groups, including a range of harm-reduction services such as syringe exchange programs, condom distribution, HIV testing, and other outreach activities.

Some comments should be made concerning the limitations of this study. First, limited coverage of HIV testing and small sampling depth in our study may have restricted the ability to detect transmission clusters. Hence, at the population level, the infection network could be wider than the one studied, with unknown transmissions occurring. Second, route of transmission data is missing for a large proportion of samples because of stigma and discrimination on the basis of real or perceived participation in key risk groups (IDUs, sexual workers or MSM) [54,55,56,57]; this missing data, to some extent, makes it difficult to understand the characteristics of HIV transmission clusters at a population level. Third, the small sample size in some countries is not fully representative of their general HIV-infected population. When combined with the large sequence number variability in each country, this may lead to bias in assessment of migration routes and detection of viral movements. Lastly, our results are limited to the *pol* region of HIV-1; the use of complete genomes may provide more accurate inference of transmission histories. Thus, some caution is advised regarding the interpretation of the findings from the present study.

## 5. Conclusions

In summary, this study based on genetic data identifies the transmission networks of two major HIV-1 subtypes in Uzbekistan, emphasizes that IDU-related and sexual exposures are involved as mechanisms of transmission in the overlapping social networks, and indicates the complexity and high burden of HIV-1 infection in the population. We also demonstrate that the existing migration flow probably contributes to an increase in the HIV infected population, and that cross-border transmissions among high-risk groups occur bi-directionally between Uzbekistan and other FSU populations. These results are of public health importance and suggest that preventive action should be reinforced in this area to improve the management of high-risk practices.

## Figures and Tables

**Figure 1 viruses-14-01675-f001:**
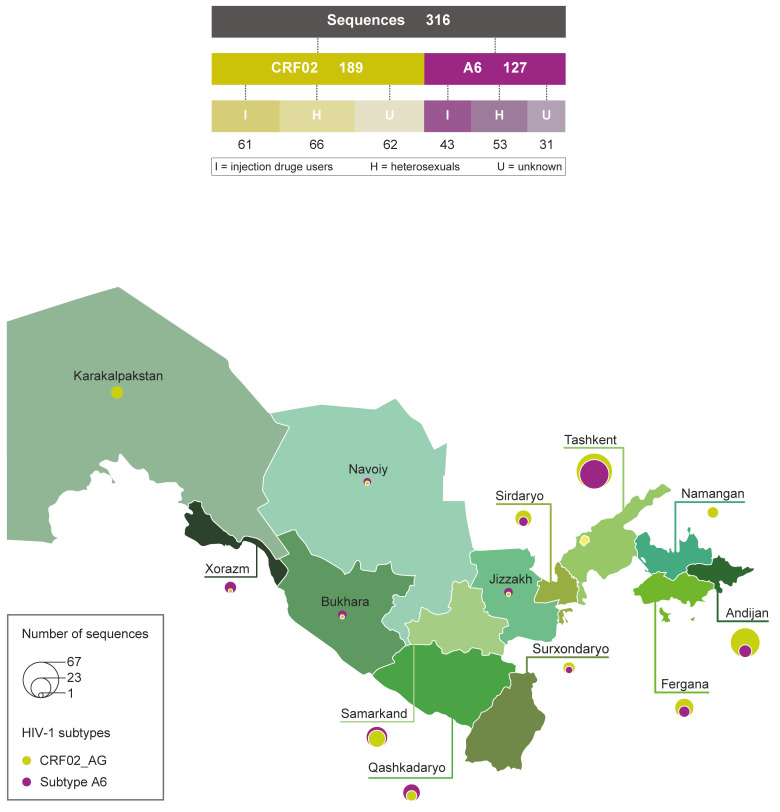
Subtypes, transmission route and geographical composition of Uzbek HIV-1 sequences enrolled in the study. Diagram (**top panel**) showing the number of two principal HIV-1 subtypes sequences and transmission route. Map of Uzbekistan (**bottom panel**) with region names and the proportions of CRF02_AG and A6 subtype sequences listed on the bubble; the circle size is proportional to the number of sequences in corresponding regions.

**Figure 2 viruses-14-01675-f002:**
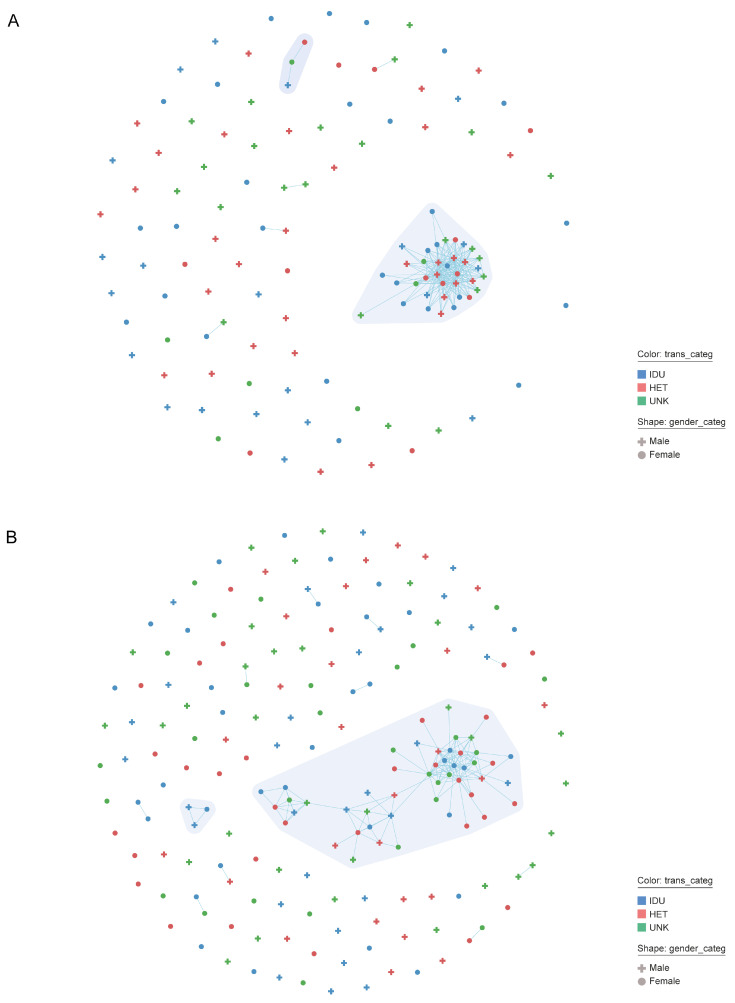
Network clustering results for the HIV-infected persons sequences in Uzbekistan generated by MicrobeTrace. The number and size of individual clusters for persons with HIV-1 CRF02_AG (**A**) and subtype A6 infection (**B**) are shown. Gender and transmission risk factors have been mapped to node shape and color, respectively, as described in the legend. Each node corresponds to an individual person, and each line represents two persons having a genetic distance 1.5% (d = 0.015). Different clusters are displayed in different polygon using a grey color gradient. Abbreviation: HET, heterosexual contacts; IDU, injection drug users; UNK, unknown.

**Figure 3 viruses-14-01675-f003:**
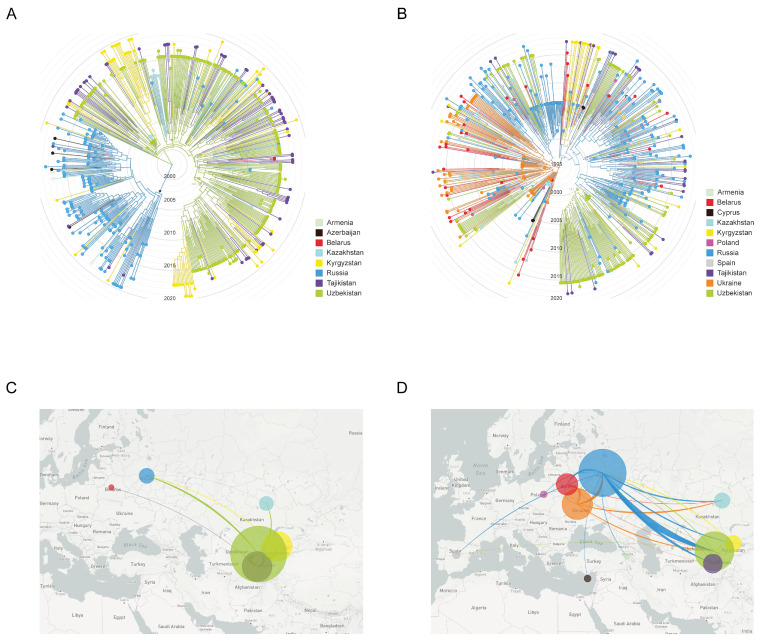
Maximum-likelihood phylogeny and inferred geographical transmission routes of Uzbek HIV-1 sequences with publicly available sequences from FSU and European countries. Shown is the analysis of HIV-1 CRF02_AG (**left panel**; **A**,**B**) and sub-subtypes A6 (**right panel**; **C**,**D**) (**A**,**B**). ML-tree shows a time calibrated phylogeny and revealing similarities of most closely related to Uzbekistan global sequences from a BLAST search at GenBank, including sequence from Caucasian and Central Asian republics, and Europe. Branches are colored by country of sampling as indicated in the legends. Branch lengths of ML tree are drawn to scale with the concentric circles indicating calendar years. The trees were rooted through HIV-1 subtype C. (**C**,**D**). The migration patterns of the HIV-1 strains in the Uzbekistan and neighboring countries. The image shows the HIV-1 transmissions (migrations) to/from different geographic areas that are colored according to the legends. The circle size is proportional to the number of sequences in corresponding regions. The graphics were generated using the auspice (https://auspice.us, accessed on 20 May 2021).

**Figure 4 viruses-14-01675-f004:**
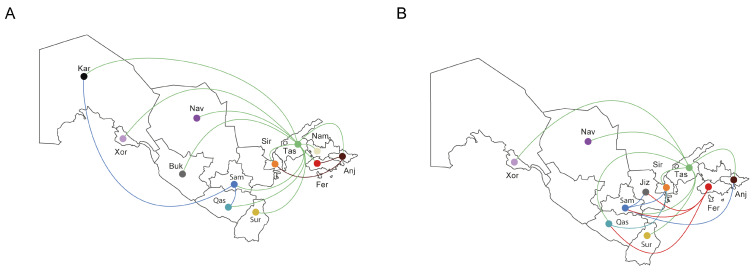
The in-country migration patterns of the HIV-1 strains (Uzbekistan). The image shows the HIV-1 sub-subtype A6 (**A**) and CRF02_AG (**B**) transmissions (migrations) to/from different geographic areas that are represented a three-letter code/color code: And, Andijan; Buk, Bukhara; Fer, Fergana; Jiz, Jizzakh; Kar, Karakalpakstan; Xor, Xorazm; Nam, Namangan; Nav, Navoiy; Qas, Qashqadaryo; Sam, Samarkand; Sir, Sirdaryo; Sur, Surxondaryo; Tas, Tashkent.

**Table 1 viruses-14-01675-t001:** Multivariable analysis of selected demographic characteristics of persons infected with HIV-1 CRF02_AG and sub-subtype A6 in Uzbekistan.

Characteristics	Estimate	Standard Error	t Value	*p* Value	Odds Ratio [95% CI]
Gender (Male vs. Female)	−0.188	0.304	−0.618	0.536	0.828 [0.454–1.509]
Age (in years)	0.021	0.016	1.310	0.191	1.022 [0.989–1.056]
Region (NER vs. Other) ^1^	0.963	0.302	3.186	0.001	2.620 [1.443–4.757]
Transmission category (IDU vs. HET)	0.254	0.297	0.853	0.394	1.289 [0.716–2.320]

^1^ NER (north-eastern region): Andijan, Fergana, Namangan and Tashkent region. Abbreviations: HET, heterosexual contacts; IDU, injection drug users; CI, confidence interval.

**Table 2 viruses-14-01675-t002:** Characterization of Uzbek HIV-1 CRF02_AG and sub-subtype A6 transmissions clusters.

	CRF02_AG			Sub-Subtype A6		
Cluster	1	2	*Dyads*	1	2	*Dyads*
Cluster (dyads) size, *N* (%)	49 (25.5)	3 (13.5)	10 (13.5)	36 (28.3)	3 (8.7)	8 (8.7)
Gender, *N* (%)						
Male	16 (32.7)	2 (66.7)	7 (35.0)	19 (52.7)	1 (33.3)	4 (50.0)
Female	33 (67.3)	1 (33.3)	13 (65.0)	17 (47.3)	2 (66.7)	4 (50.0)
Transmission route, *N* (%)						
HET	15 (30.6)	3 (100.0)	11 (55.0)	14 (38.9)	1 (33.4)	2 (25.0)
IDU	19 (38.8)	0	6 (30.0)	13 (36.1)	1 (33.3)	2 (25.0)
UNK	15 (30.6)	0	3 (15.0)	9 (25.0)	1 (33.3)	4 (50.0)
Residence or region of sampling	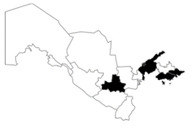	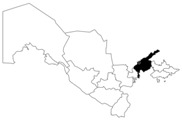	N/a	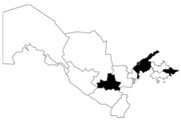	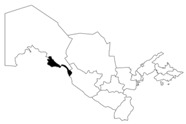	N/a
Diagnosis date	2013–2016	2015	2002–2015	2002–2016	2015	2013–2015
Likely phylogenetic origin (country)	Uzbekistan	Uzbekistan	N/a	Russia	Russia	N/a
TMRCA (95% CI)	02-03-2003 [28-04-2002, 12-12-2004]	27-12-2005 [20-03-2004, 30-06-2008]	N/a	08-02-2000 [11-02-1999, 31-07-2001]	07-03-2009 [12-09-2007, 08-01-2011]	N/a

Abbreviations: HET, heterosexual contacts; IDU, injection drug users; UNK, unknown; TMRCA, time to the most recent common ancestor; CI, confidence interval; N/a, not assessed.

**Table 3 viruses-14-01675-t003:** Assortativity for the HIV-1 CRF02_AG and sub-subtype A6 clusters in Uzbekistan by gender, region and transmission category.

	CRF063_02A		Sub-Subtype A6	
Cluster	1	Total ^1^	1	Total
Number of Nodes, *N*	49	72	36	47
Assortativity, *r*				
Region	0.29	0.30	0.27	0.21
Sex	0.03	0.03	−0.08	−0.09
Transmission route	−0.04	<0.01	−0.18	−0.06

^1^ Estimates across the clusters and dyads in aggregate.

## Data Availability

Not applicable.

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
