# Peer review of "Identifying HIV-1 Transmission Clusters in Uzbekistan through Analysis of Molecular Surveillance Data"

_viruses, 2022, doi:10.3390/v14081675_

Round 1
Reviewer 1 Report
When reading the manuscript, two remarks arose.
1. On page 4 (lines 317-318) of the Discussion section, the authors state that a significant proportion of non-clustered sequences are associated wıth “dead-end transmission events”. However, even in this case, the sequences must cluster forming dyads. According to the reviewer’s opinion, low clustering is more related to the low sampling density. It should be taken into account that the size of the analyzed sample in the work was less than 1000 sequences, while the total number of HIV-infected people in Uzbekistan is 58,000 people. The importance of sampling density has been well demonstrated (Novitsky V, Moyo S, Lei Q, DeGruttola V, Essex M. Impact of sampling density on the extent of HIV clustering. AIDS Res Hum Retroviruses. 2014 Dec;30(12):1226 -35 doi: 10.1089/aid.2014.0173 PMID: 25275430; PMCID: PMC4250956).
2. Since the amount of the analyzed samples is not very large, it is recommended to analyze how well the studied cohort reflects the entire population of HIV-infected people in Uzbekistan. To do this, can be recommended adding another column in Figure S1 and present the epidemiological characteristics of the population of HIV-infected citizens of Uzbekistan. The ability to compare the characteristics of analyzed cohort and the entire population of HIV-positive people in country in one table will make it possible to visualize the bias of the cohort.
Reviewer 2 Report
The study by Lebedev et al investigates the epidemiological history of the HIV-1 subtype A6 and CRF02_AG in Uzbekistan by analysing 316 pol sequences sampled between 2010 – 2016. Cluster analysis revealed one major cluster for each subtype indicative of a centralized epidemic in the country. The authors then combined their sequence data with publicly available sequence data from other countries and perfumed phylogenetic analysis. This supported the finding with little mixing of transmission across states. The study is overall well presented with sound methods and results. However, there some aspects which need more clarification. My main critique is the discussion focuses heavily on the across country transmission and linkage to migration and sex work and individual behaviour. However, the study presented does not have enough data to support these assumptions. I understand ‘country of birth / migrant or not’ was not included as a demographic factor and most sequence data used was obtained from infections among locals. I think the authors should discuss more the geographical difference in subtype distribution they found. What are the epidemiological reasons for this? Limited sampling is a definite limitation for the study. The authors used 316 sequences across 6 years, it is not clear what proportion of total new infection that represents. Also, sampling bias is always a limiting factor when using global data. While the authors do mention this in limitation paragraph, it should also be noted in the discussion, particular with links to transmission routes across countries.
Minor comments
Line 79 ff. ‘additional 52 sequences’ I assume these are used to support the subtype classification as this is mentioned in the following paragraph. Thus, I think this sentence should be moved to that paragraph (line 105).
Line 84. 316 sequences were used but it says ‘Four sequences were excluded’. However, the total of 316 were used for MicrobeTrace and Nexstrain. Please clarify
It reads ‘including all 316 ‘Uzbek’ pol sequences’ but in the paragraph above it says 4 sequences were not included in the analysis (line 84). Please clarify
Line 112. ‘to cluster selection in ML tree’ . I don’t understand what the authors are trying to say here. From my understanding the MicrobeTrace method uses genetic distance only to determine networks. Was a maximum likelihood tree also used for cluster analysis? Please clarify.
Line 141. Region of origin. I think region of residence is more appropriate. Nevertheless, the terminology should be consistent throughout the manuscript.
Is MSM not captured as a demographic? About 1/3 of sequences are linked to a ‘unknown’ risk factor. A this is a large proportion has it been included in the statistics?
Lines 143 ff. Why where these specific regions combined to the North Eastern Region. Is this a commonly used sub-region of the country? Are there demographic differences between this sub-region and other parts of the country?
Line 160. ‘5 transmission clusters’ but only 4 are shown in the figure and table.
Line 172. What do the authors mean by ‘female had a significant presence’ . Does this mean infections among women were more likely to be in clusters than not? This is not evident from the data shown.
Line 174 ff. I don’t understand this sentence.
Line 194. (Figure legend) ‘mother-to-child’ is not shown in the figure.
Line 270. (Figure legend). It reads ‘Branches are colored by country of birth’. Should this be country of sampling?
Line 330. I think ‘support’ is a better word here than ‘confirms’.
